# Effects of Natural Habitat and Season on Cursorial Spider Assemblages in Mediterranean Vineyards

**DOI:** 10.3390/insects14100782

**Published:** 2023-09-25

**Authors:** Zeana Ganem, Marco Ferrante, Yael Lubin, Igor Armiach Steinpress, Moshe Gish, Rakefet Sharon, Ally R. Harari, Tamar Keasar, Efrat Gavish-Regev

**Affiliations:** 1The National Natural History Collections, The Hebrew University of Jerusalem, Jerusalem 9190401, Israel; zeana.ganem@mail.huji.ac.il (Z.G.); igor.armiach@mail.huji.ac.il (I.A.S.); 2School of Environmental Sciences, University of Haifa, Haifa 3103301, Israel; mgish@univ.haifa.ac.il; 3Centre for Ecology, Evolution and Environmental Changes (cE3c), Azorean Biodiversity Group, Faculty of Agriculture and Environment, University of the Azores, Angra do Heroísmo, 9700-042 Azores, Portugal; marco.ferrante@uni-goettingen.de; 4Mitrani Department of Desert Ecology, Ben-Gurion University of the Negev, Midreshet Ben-Gurion 8499000, Israel; 5Northern R&D, MIGAL-Galilee Technology Center and Tel-Hai Academic College, Qiryat Shemona 1220800, Israel; rakefets@migal.org.il; 6Department of Entomology, Agricultural Research Organization, The Volcani Center, Bet Dagan 7505101, Israel; aharari@volcani.agri.gov.il; 7Department of Biology and the Environment, University of Haifa—Oranim, Tivon 3600600, Israel; tkeasar@gmail.com

**Keywords:** agroecosystem, Araneae, Levant, species composition

## Abstract

**Simple Summary:**

Spiders are potential natural enemies of insect pests in many crops, and their species composition in the crop may be influenced by nearby natural habitats. Here, we examined the effects of the habitat type (different sampling positions within the vineyard and in the nearby natural habitat) on spider assemblages in vineyards. Spider species richness, assemblage composition, and diversity were evaluated by means of pitfall traps in early and late summer, in three commercial vineyards and their adjacent natural habitats in a Mediterranean landscape in northern Israel. We collected 688 spiders, belonging to 25 families and 61 species and morphospecies. Spider richness differed in the two seasons; more species were documented in early summer (47) than in late summer (33). The natural habitat had the highest species richness, with 34 species, while three vineyard positions were inhabited by only 27–31 species each. The natural habitat assemblage differed from the vineyard assemblages, with 15 species that were found only in the natural habitat, yet 11 species were shared by both the natural habitat and all vineyard positions. Both season (early vs. late in the cropping season) and the habitat (vineyard vs. natural) affected the spider assemblage composition. The study documents the large diversity of spiders in a Mediterranean vineyard agroecosystem. The information that we provide here is critical in assessing the potential for conservation biocontrol, where natural habitats may be a source of natural enemies for nearby vineyards.

**Abstract:**

Natural habitats adjacent to vineyards are presumed to have a positive effect on the diversity of natural enemies within the vineyards. However, these habitats differ in vegetation structure and seasonal phenology and in turn could affect the species composition of natural enemies. Here, we compared the species richness and diversity and the composition of spider assemblages in several locations within three commercial vineyards and the nearby natural habitats in a Mediterranean landscape in northern Israel. We sampled spiders by means of pitfall traps in early and in late summer. Both the time in the season and the habitat (natural versus vineyard) affected spider species richness and diversity. More species were found in early summer (47) than in late summer (33), and more occurred in the natural habitat (34 species) than in the vineyards (27–31 species). Fifteen species were found exclusively in the natural habitat, and only 11 species were shared by the vineyards and natural habitat, four of which were the most abundant and geographically widely distributed species in the samples. In late summer, spider diversity in the natural habitat was higher than within the vineyards: the spider assemblages in the vineyards became dominated by a few species late in the crop season, while those of the natural habitat remained stable. Overall, the natural habitat differed in assemblage composition from all within-vineyard locations, while the three locations within the vineyard did not differ significantly in assemblage composition. Season (early vs. late summer), however, significantly affected the spider assemblage composition. This study documents the large diversity of spiders in a local Mediterranean vineyard agroecosystem. Over 60% of the known spider families in the region occurred in our samples, highlighting the importance of this agroecosystem for spider diversity and the potential for conservation biocontrol, where natural habitats may be a source of natural enemies for nearby vineyards.

## 1. Introduction

The biological control of insect pests in agroecosystems by means of arthropod predators is an ecosystem service valued at >USD 400 billion/year globally [1]. Natural and semi-natural habitats within the agricultural landscape can potentially aid biological control in crops [2]. Such habitats tend to be less disturbed and more heterogeneous [3,4,5] and contain diverse plant communities [6] and more ground cover [7] than the crop. Moreover, natural and semi-natural habitats can have synergistic, positive effects with the implementation of agri-environmental schemes (e.g., establishment of flower strips or hedges), resulting in better conditions for ecosystem service providers [8]. 

Spiders are important generalist predators in agroecosystems [9]. Worldwide, they are estimated to kill 400–800 million tons of prey every year, much of this likely in crops [10]. Their diverse hunting strategies [11] and ability to establish populations early in the crop season make them efficient biological control agents of pests [12,13,14]. Simplification of the agricultural landscape through habitat homogenization has led to a decline in biodiversity worldwide, jeopardizing ecosystem services on which farming systems depend [4]. It is suggested that preserving natural and semi-natural habitats and their vegetation at the habitat and landscape levels will enhance the assemblage composition of spiders, as well as other beneficial groups, and increase their role in biological pest control [15,16]. A diverse spider assemblage is expected to increase their effectiveness as predators in the crop as species-rich assemblages benefit from functional complementarity [17]. 

Spiders can disperse from natural and semi-natural habitats into crop fields over short distances by walking, or over longer distances by ballooning [18,19]. Dispersal by walking is more common in ground-dwelling spiders. Such movements can occur even in low-productivity ecosystems such as deserts; for instance, more than half of the species in wheat fields in the Negev desert (Israel) immigrated from the surrounding semi-natural habitats [13,20]. However, many studies indicate that spider species show a high degree of habitat specialization, which may limit dispersal from adjacent natural and semi-natural habitats to crop fields [21,22,23]. In addition, widely distributed agrobiont species are often typical of disturbed habitats and occur less commonly or not at all in adjacent natural habitats [9]. 

Seasonal changes in both the crop and natural habitat can interact with each other, affecting the spider assemblage composition. This may be due to biotic factors, such as changes in the availability of resources, as well as abiotic factors, such as thermal tolerances. For instance, the diversity and abundance of spider assemblages in four natural habitats in the Negev desert (Israel) varied dramatically between the warm and cold seasons [24]. At a field scale, the crop phenology and the season affected the activity patterns of different spider taxa [13,25,26] and, hence, the assemblage composition. Finally, the crop type is a strong determinant of the spider fauna, e.g., perennial crops are likely to have taxonomically and functionally diverse spider assemblages due to their phenology and complex structure (see [23,27]).

Vineyards are a perennial crop and thus spiders may persist in the vineyard throughout the seasons. Their abundance and species composition, however, can vary with crop phenology. In South African vineyards, spiders were increasingly abundant in the growing season (spring), thereby likely contributing to pest suppression early in the season [28]. Vineyards are often under intensive management, including pesticide application and the removal of weeds between vine rows [29,30]. Despite intensive management, habitats surrounding the vineyards, as well as the plant cover within the vineyard, can influence the local spider assemblages. In Mediterranean vineyards in northern Italy, heterogeneous landscapes with nearby woody natural habitats favored the occurrence of ambush hunters, while sheet-web spiders benefitted from less diverse landscapes [31]. Spider densities were positively affected by nearby semi-natural habitats in Germany, but organic vineyards only moderately increased the diversity of spiders compared to conventionally managed ones [32]. In South African vineyards, spider diversity and abundance were enhanced by ground cover within the vineyard [33], by structurally complex natural habitats surrounding the vineyards [34], and by a combination of management regime type and nearby natural habitats [28]. By contrast, there was little effect of the surrounding landscape (woody or grassland) on the abundance of different spider families in Australian vineyards [35] and, similarly, of within-vineyard vegetation on spider abundance and species richness [36]. Finally, the spider assemblage composition can differ among even close locations and may be quite distinct from that of nearby habitats [23,37]. These diverse results suggest that nearby habitats, along with the season, management regimes, and regional variables, may be involved in complex ways in determining the spider assemblages within vineyards. 

Our study was conducted in a previously poorly investigated vineyard agroecosystem in the Eastern Mediterranean. We aimed to determine the degree of similarity or difference in the species richness, diversity, and composition of spider assemblages in the vineyards and adjacent natural woodland habitats in this agroecosystem and the effect of the season within the crop phenology on these assemblages. We evaluated the spider assemblage composition and species richness and diversity in three commercial vineyards and their adjacent natural habitats in a Mediterranean landscape in northern Israel. The vineyards were relatively small (~6 ha) and embedded in a Mediterranean woodland landscape, consisting of patches of herbaceous plants, small trees (oaks and *Pistacia*) and shrubs. We sampled the ground-active spiders at the beginning of the grape-growing season and before the time of fruit harvest. 

We hypothesized that the spider assemblages of early and late summer would differ, as well as those in different habitats. We predicted that the spider species richness would be greatest in the natural woodland habitat and least in locations distant from the edge of the vineyard. This is because the vegetation diversity of the natural habitat is greater than in the crop, and because the vineyards were subjected to the application of herbicides to remove weeds and thus had lower vegetation cover, as well as higher levels of pesticides, compared to the adjacent natural habitats [38]. Moreover, we expected the natural habitats to host habitat-specialist species that cannot tolerate highly disturbed habitats such as managed crops. For similar reasons, we predicted greater similarity between the assemblage compositions at different locations within the vineyards than between the vineyards and the adjacent natural habitats. Moreover, we expected this pattern to be more evident in late summer, before harvest time, when the differences in vegetation between the natural habitat and the vineyard are greater. 

We hereby add new information on the assemblages of ground spiders in the Levantine vineyard agroecosystem, and on the environmental factors that affect these assemblages. This information is critical in assessing the potential for conservation biocontrol, where natural habitats may provide a source of natural enemies for nearby vineyards.

## 2. Materials and Methods

### 2.1. Study Sites

The Qedesh Valley, Upper Galilee, Israel has a Mediterranean climate with a distinct rainy cold season (October–March) and a dry hot season (April–September), with an average of 600 mm annual rainfall. In this region, three Cabernet Sauvignon wine-producing vineyard sites (Dishon, Zar’it, and Yiftah) were sampled twice during 2014, in early summer (June) after vine flowering, and in late summer (August) before harvest time. The vineyards were under integrated pest management (Appendix A), and all received insecticide and herbicide applications and were drip-irrigated from June until August (see [38,39] for details). Each site had vineyard bordering a natural habitat on at least one side, separated by a dirt track (Dishon ~5 m wide, Zar’it and Yiftah ~7 m wide). The natural vegetation was a mix of herbaceous plants, shrubs, and oak and *Pistacia* trees. All vineyards had sparse vegetation cover (<2%) between the vine rows due to two–three herbicide applications in April and May (Appendix A). The annual vegetation cover in the natural habitat was highest in spring and early summer (first sampling) and gradually declined towards mid-summer as annual plants dried.

### 2.2. Spider Sampling

In each of the three sites, we installed twenty-four pitfall traps along two transects, at four positions (habitats) at fixed distances from the border of the vineyard with the natural habitat, for a total of 72 pitfall traps. The positions were (1) 30–50 m into the natural habitat from the adjacent focal vineyard border (natural habitat—NH); (2) 10 m into the focal vineyard, at the border with the natural habitat (border—BO); (3) 60 m from the border with the natural habitat into the focal vineyard (center—CE); and (4) 100 m from the border with the natural habitat into the focal vineyard, close to the neighboring vineyard (neighbor vineyard—VV; Figure 1). Distances for treatments NH and CE were chosen to avoid edge effects, while those for treatments BO and VV were chosen to sample the two edge positions. There were three pitfall traps at each position on each of two transect lines; thus, there were a total of six traps per position in each site. The two transect lines were separated by >10 m. Within the vineyard, traps were placed beneath the vines, along a single row of vines parallel to the border with the natural habitat. The traps were ~3 m from one other and separated by at least one vine plant. In the natural habitat, traps were similarly placed parallel to the vineyard border. Each pitfall trap consisted of two plastic cups placed one within the other, 10 cm deep with an opening diameter of 9 cm, and dug into the soil so that the opening of the outer cup was level with the soil surface. Each trap contained a 60 mL mixture of water (40%), absolute ethanol (20%), and propylene glycol (40%). The traps were covered with a net to prevent the capture of small vertebrates and were left for one month for each sampling event. 

Wild boars damaged 10 traps during the first sampling (2 NH, 4 BO, 4 VV) and 12 traps during the second sampling (8 NH, 1 BO, 3 CE); the damaged traps were excluded from the analyses. Spiders were identified using identification keys from the Araneae—Spiders of Europe website [40] and the Field Guide to the Spiders of Britain and Northern Europe [41]. Juvenile spiders were identified to family or genus level, when possible, while adults were identified to species or morphospecies. Consequently, we analyzed two datasets: one that included all spiders identified to the family level (*n* = 632 individuals), and a subset of these data that included all spiders identified to the species or morphospecies level (*n* = 366 individuals). All spiders were deposited in the Israel National Arachnid Collection, at the National Natural History Collections, The Hebrew University of Jerusalem.

### 2.3. Data Analysis

Rank abundance curves were produced for the two datasets: families, and species and morphospecies (hereafter referred to as “species dataset”). Using both datasets, we visualized the effect of the sampling season (early and late summer) and the different trap positions in the vineyard (NH, BO, CE, VV) on spider species and family composition. We statistically analyzed these effects using direct constrained ordination. We used both redundancy analysis (RDA) ordination and partial RDA. In the RDA, we tested together two variables, season (two levels) and position (habitat—four levels), while the third variable, site (three levels: Dishon, Yiftah, Zar’it), was set as a co-variable in all our analyses. In Canoco, for categorical variables that have more than two levels, the levels are tested as separate categories (variables). We therefore used forward selection to test the position levels in RDA and included in the model only significant variable levels. In the partial RDA, we tested one variable serving as the main effect, while the others (site, and either season or position) were set as co-variables [42,43]. In both analyses, we used 9999 unrestricted permutations under a full model. Species overlaps between the sampling positions were also visualized in a Venn diagram.

Species composition in the four sampling positions (habitats) in late spring and late summer was further characterized using diversity profiles as Rényi curves Equation (1), since single diversity indices may rank communities differently [44,45]. The following equation Equation (1) expresses the relationship between the number of species and their relative abundances in a community.
(1)$$H=\frac{1}{1-a}log\sum_{i=1}^{q}p_{i}^{a}$$
where *q* is the number of species and *p* is the relative frequency of species in the sample. The constant value *α* can take any positive values between 0 and *α*→∞. When *α* is = 0, *H* is equal to the logarithm of the species richness; when *α*→1, *α* = 2, and *α*→∞, *H* is related to the Shannon diversity, Simpson’s diversity, and inverse Berger–Parker dominance indices, respectively (i.e., the weight of dominant species increases with increasing values of *α*). When the diversity profile of one species assemblage is above that of another assemblage, the first is unequivocally more diverse than the other (i.e., higher diversity for all values of *α*), while, when the curves cross, the two assemblages cannot be unequivocally ranked by diversity.

All statistical analyses were performed using Canoco for the ordinations [42,43] and R version 4.1.1 [46] and the package vegan [47] for the other analyses.

## 3. Results

### 3.1. Spider Abundance

Overall, 688 spider individuals were collected, belonging to 25 families (Figure 2a, Appendix A and Appendix B). Fifty-six individuals were not assigned to a family due to their poor preservation. Sixty-one species and morphospecies were identified (366 individuals in total), representing 53% of the individuals collected. Thirty-three species (5.6% of all trapped individuals) were singletons and doubletons (Figure 2a, Appendix A). The most abundant spider families were Gnaphosidae, Zodariidae, Salticidae, Lycosidae, Philodromidae, and Linyphiidae, which accounted for 85.4% of the spiders caught that were identified to the family level. The four most abundant species were *Pterotricha levantina* Levy, 1995 (Gnaphosidae, 51 individuals), *Thanatus vulgaris* Simon, 1870 (Philodromidae, 36 individuals), *Hogna graeca* (Roewer, 1951) (Lycosidae, 28 individuals), and *Zodarion lutipes* (O. Pickard-Cambridge, 1872) (Zodariidae, 20 individuals).

### 3.2. Spider Richness

The highest species richness was found in the natural habitat (NH, 34 species, of which 15 were found only in this position), followed by the position between the focal vineyard and an adjacent one (VV, 31), the border (BO, 29), and the vineyard center (CE, 27). Eleven species were found in all four positions and only four species were exclusively found in the vineyard center: *Pterotricha cambridgei*, *P. parasyriaca*, *Ozyptila* sp., and one unidentified linyphiid (Figure 3). Spider diversity differed greatly over the season; more species were documented in early summer (47) than in late summer (33). The six species with more than two individuals collected that were found only in June were *Zelotes cf. zin* (11 individuals, all sites), *Pritha albimaculata* (O. Pickard-Cambridge, 1872) (8 individuals, Zar’it and Dishon), *Heser aradensis* (Levy, 1998) (5 individuals, Yiftah and Dishon), *Prochora lycosiformis* (O. Pickard-Cambridge, 1872) (5 individuals, Yiftah and Dishon), *Oecobius* sp. (4 individuals, Yiftah only), and *Palaestina eremica* Levy, 1992 (3 individuals, Dishon only). The three species that were found only in August were *Micaria ignea* O. Pickard-Cambridge, 1872 (17 individuals, Zar’it only), *Civizelotes solstitialis* (Levy, 1998) (6 individuals, Dishon and Zar’it), and *Singa neta* (O. Pickard-Cambridge, 1872) (3 individuals, Zar’it and Yiftah).

### 3.3. Spider Species and Family Composition

The spider species composition was significantly affected by the sampling season (partial RDA; season, df = 1, F = 2.2, *p* = 0.008, Figure 4, constrained axis 1), which explained 11.5% of the variance in species composition. Position (habitat) explained an additional 8.3% of the variance in the species composition, but this effect was not significant (partial RDA; position, df = 3, natural habitat F = 1.7, *p* = 0.078; Figure 5, constrained axis 1). Figure 4 shows the distribution of the 12 species with the highest fit to the ordination plot across the sampling seasons. Axis 1 represents the season, while the other axes are not constrained. Axis 2 may represent a combination of other non-constrained variables such as the position (habitat), the specific vineyard, or other variables that we did not measure. Figure 5 shows the distribution of all species across positions. Axis 1 is constrained and represents the position (habitat), while the other axes are not constrained. Axis 2 probably represents a combination of the season and other non-constrained variables, such as the specific vineyard, or other variables not measured. The three most abundant species in our study that were documented almost equally from all four positions were *Pterotricha levantina* (Gnaphosidae, 51 individuals), *Hogna graeca* (Lycosidae, 28 individuals), and *Zodarion lutipes* (Zodariidae, 20 individuals). *Micaria ignea* (Gnaphosidae, 17 individuals) was mainly recorded in the border between the natural habitat and the vineyard, while *Thanatus vulgaris* (Philodromidae, 36 individuals) was mainly recorded in the center of the focal vineyard, and *Aelurillus politiventris* (O. Pickard-Cambridge, 1872) was recorded mainly in the natural habitat (Figure 5).

The spider family composition was significantly affected by the sampling season (partial RDA; season, df = 1, F = 1.8, *p* = 0.005, Figure 6, axis 1), explaining ca. 10% of the variance, and also by the trap position (habitat) (partial RDA; position, df = 3, natural habitat F = 1.8, *p* = 0.018, Figure 6, axis 2), explaining an additional 9.1% of the variance. Figure 6 shows an ordination (RDA) for the 25 families and the significant environmental variables (season and position, first and second axes, respectively), with the site as a co-variable. Four spider families, namely Idiopidae, Oecobiidae, Palpimanidae, and Pholcidae, were documented solely in the natural habitat. Figure 7 presents the 24 samples (4 positions × 3 sites × 2 seasons) by their positions on the ordination plot and shows that within-vineyard assemblages (CE and VV) were less variable in their spider species composition (smaller areas of envelopes) than the assemblages of the natural habitat and the border between the vineyard and natural habitats.

### 3.4. Spider Diversity

The Renyi diversity profiles of the spider assemblages in the four positions differed in the two sampling seasons. In early summer, the natural habitat showed the highest species richness and slightly higher Shannon diversity and Simpson’s diversity indices, while its inverse Berger–Parker index dropped as α increased; however, there were no major differences in diversity among the vineyard and natural habitat assemblages (Figure 8). In late summer, however, the spider assemblage in the natural habitat was more diverse, but with lower species richness, than those in the vineyard positions. The spider assemblage at the border with the natural habitat was unequivocally the least diverse. No large differences were observed between the spider assemblages in the center of the vineyard and near the neighboring vineyard both in early and late summer (Figure 8).

## 4. Discussion

The aim of this study was to evaluate the effect of season and habitat on spider species richness, diversity, and assemblage composition, thereby contributing to the knowledge of spider diversity in Mediterranean vineyard agroecosystems. Overall, we found 61 species/morphospecies from 25 families in this agroecosystem, which constitutes over 60% of the spider families known for this region [48]. This is similar to the spider species richness (62 species from 17 families) that was found in Central European vineyards [32] and higher than the richness recorded in South African vineyards (45 species from 16 families [28]). Our results indicate that even in agricultural landscapes, vineyard spider assemblages can be diverse, and that nearby natural habitats contribute greatly to the overall species richness. We did not investigate the mechanisms through which natural habitats support ground-active spider assemblages, but it is possible that species that were found exclusively in this habitat benefit from the less severe micro-climatic conditions (e.g., higher humidity and lower temperatures compared to Mediterranean vineyards).

Effects of season: In agreement with our hypothesis, we found a large effect of the season on the assemblage composition, as seen in the ordinations (Figure 4 and Figure 6). The spider assemblage composition at both the species and the family level differed between the seasons, with 19 shared species between the seasons, 47 species that were found only in early summer, and 33 only in late summer. The level of identification of the spiders to species allowed us to discover some species replacements over the season. These patterns may represent species’ natural history, development, and habitat use. For example, the ground-dwelling gnaphosid *Pterotricha levantina*, which was the most abundant species in our study, was documented in higher abundance in late summer and was probably replacing the ground-dwelling lycosid *Pardosa subsordidatula*, which was most abundant in early summer. Both species are nocturnally active hunters, but *Pterotricha* is more typical of xeric conditions than *Pardosa*, and this may explain its abundance in late summer (pers. obs., E.G.-R.).

The diversity curves showed no major differences between the four trap positions (habitats) in early summer, indicating similar evenness of species abundances, while the spider assemblage in the natural habitat was clearly the more diverse in the late summer. This pattern was likely due to the fact that, in the late summer, only a few species dominated the spider assemblage in the vineyards, while the spider assemblage in the natural habitat had higher evenness and was likely more stable.

Effects of habitat: We predicted that spider richness would be greatest in the natural habitat. This, indeed, was the case. The natural habitat had 34 species, i.e., over half of the species/morphospecies found altogether, in comparison with three vineyard positions (27–31 species). A possible explanation for this pattern is the greater vegetation diversity of the natural habitat compared to the crop, partly due to the application of herbicides to remove weeds within the vineyards [38].

The natural habitats hosted 15 habitat-specialist species that likely cannot tolerate disturbed habitats such as managed crops. Families specific to the natural habitat were Idiopidae, Oecobiidae, Palpimanidae, and Pholcidae. These included some species with specialized microhabitats and hunting tactics, such as *Titanidiops syriacus* (O. Pickard-Cambridge, 1870) (trap-door burrow dwelling), *Palpimanus sogdianus* Charitonov, 1946 (a member of a family of specialist spider predators; [49]), and *Trygetus sexoculatus* (O. Pickard-Cambridge, 1872) (presumed to feed solely on ants; [50]). Gnaphosidae, one of the most abundant and species-rich families in the region [48] and in our samples, was strongly associated with the natural habitat in late summer (Figure 6). Nevertheless, some gnaphosid genera, such as *Micaria* and *Pterotricha* (which we recorded in this study), may be able to disperse into the vineyard habitat, as is known to occur between natural habitats and wheat fields [13]. By contrast, small sheetweb-building spiders of the family Linyphiidae, which were associated with the vineyard habitats, are known to be common and abundant agrobionts in various crops in Israel and in Europe [13]. Similarly, another vineyard-typical group, Titanoecidae, occurs commonly in crops in Israel (pers. obs., E.G-R and Y.L.).

As predicted, we also found greater similarity between the assemblages in positions inside vineyards (vineyard border, center, and near the neighbor vineyard) than between the vineyard positions and the adjacent natural habitats. The similarity between spider assemblages within the vineyards was particularly marked in early summer. In late summer, the spider assemblage at the border between the vineyard and the natural habitat was the least diverse, being dominated by two species of ground-running spiders (Gnaphosidae), *Pterotricha levantina* and *Micaria ignea*, the latter found exclusively in late summer and mainly in the border habitat. For these species, the ecotone between the vineyard and natural habitat may be the most suitable habitat because it provides resources from both neighboring habitats (see, e.g., [51]). An alternative explanation is that these genera are not related to a particular habitat and use ecotones to disperse in the landscape.

Of the eleven species that were found in both the vineyard and the natural habitat, four were the most abundant species in our study system, namely *P. levantina*, *Hogna graeca* (Lycosidae), *Zodarion lutipes* (Zodariidae), and *Thanatus vulgaris* (Philodromidae). These species are widespread and likely have broad habitat tolerance. *Thanatus vulgaris*, for example, has a nearly worldwide geographic distribution [52] and in Israel is found in both natural and agricultural habitats [20], while *H. graeca* is found across habitats in Israel between June and December (pers. obs., I.A.S.). No species were shared between the natural habitat and the position near the neighboring vineyard (which was also the furthest from the natural habitat), suggesting some separation of the natural habitat and the within-vineyard spider assemblages.

Our findings agree with results from other studies of spider assemblages in natural and agricultural (both annual and perennial) habitats in other parts of Israel [13,24,48] and indicate that natural habitats neighboring vineyards support spider species that are not found elsewhere in the agricultural landscape. This underlines the irreplaceable role of natural habitats to preserve highly diverse spider communities, which in turn may translate into enhanced natural pest control. Our findings are also in line with recommendations for more sustainable agriculture that emphasize habitat diversification in agricultural landscapes [53]. The establishment of non-crop habitats (e.g., [54]), as well as the conservation of existing natural and semi-natural habitats (e.g., [55]), can contribute to higher levels of biodiversity and ecosystem services and thus increase the resilience of agricultural systems.

## Figures and Tables

**Figure 1 insects-14-00782-f001:**
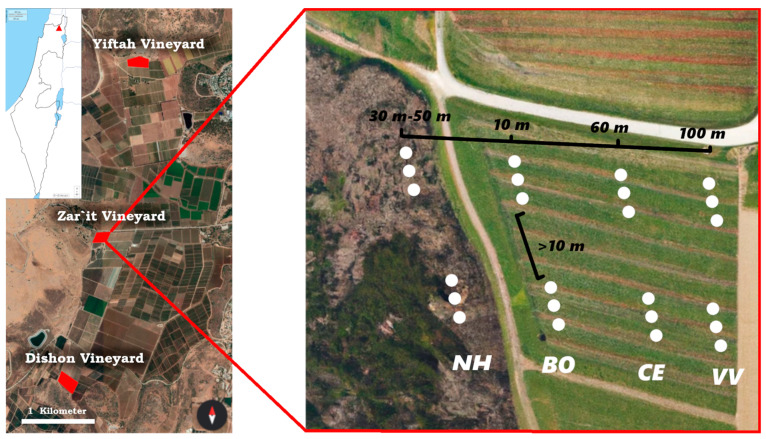
**On the left**: Map of the three study sites. The insert shows the location in Israel (red triangle). **On the right**: The four positions (habitats) sampled with pitfall traps in early and late summer 2014. From the outermost to the innermost, these are the “NH” (natural habitat, 30–50 m into the natural habitat from the vineyard border), “BO” (border, 10 m into the vineyard), “CE” (center, 60 m into the vineyard), and “VV” (close to the neighboring vineyard, 100 m into the focal vineyard near the neighboring vineyard).

**Figure 2 insects-14-00782-f002:**
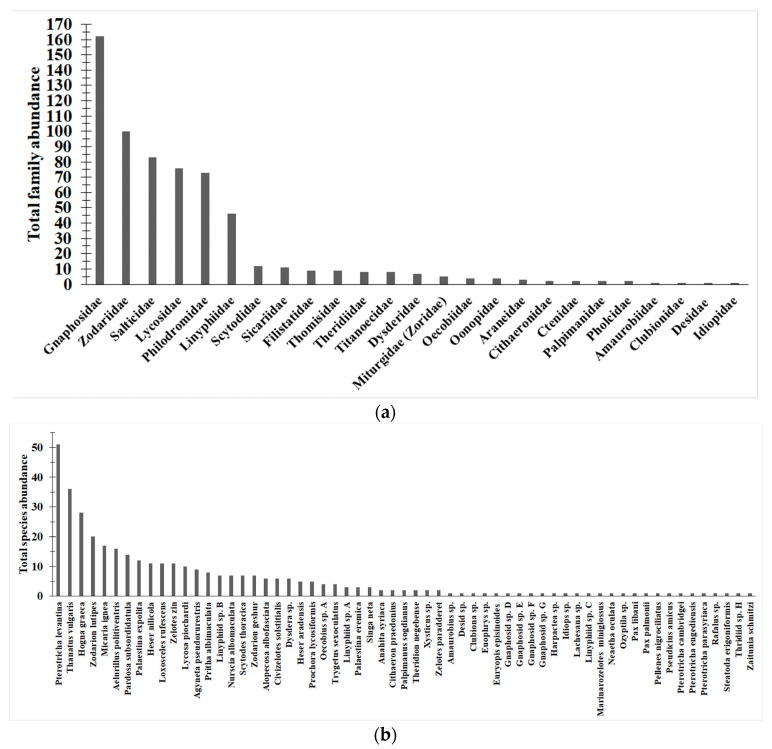
Rank abundance of Emeq Qedesh spiders. All identified spiders, in all sites, positions, and seasons, were combined. (**a**) Family level. (**b**) Species level (including morphospecies).

**Figure 3 insects-14-00782-f003:**
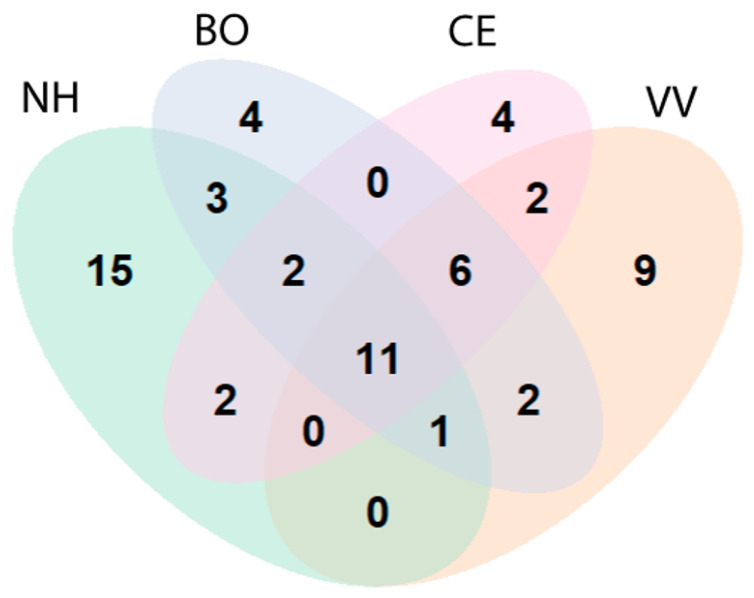
Venn diagram of the number of spider species found at each of the four positions (habitats) and the species overlap between the positions.

**Figure 4 insects-14-00782-f004:**
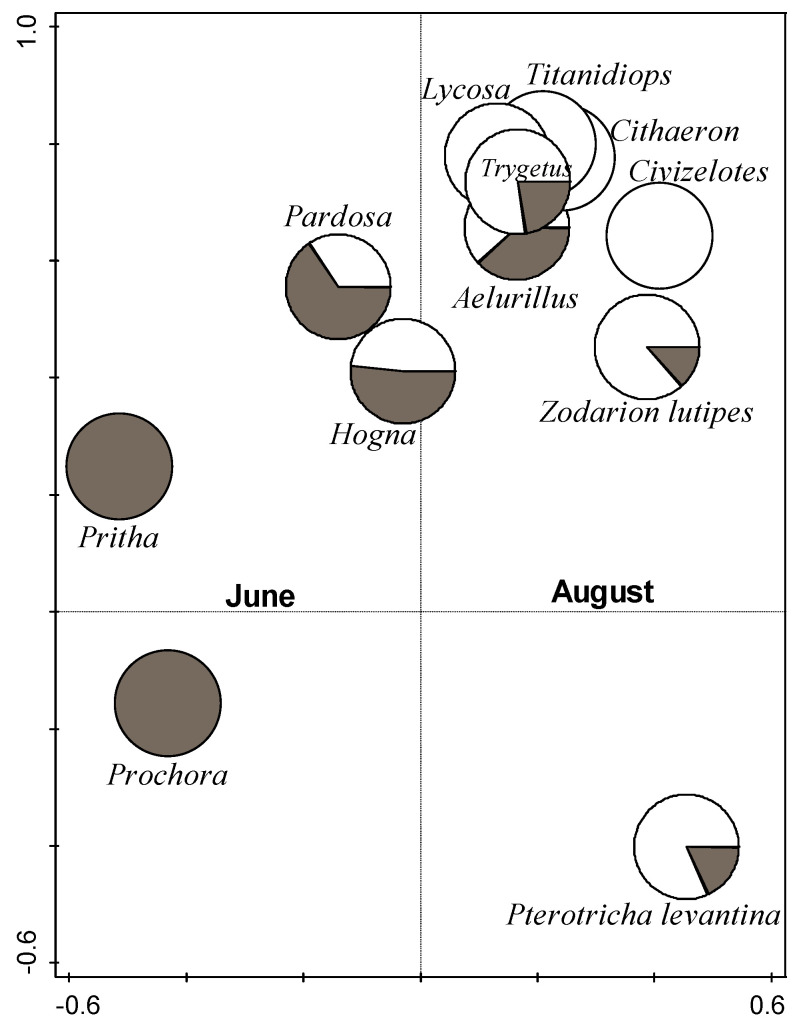
Constrained partial RDA ordination. The 12 species with the highest fit to the ordination plot are presented as pie diagrams. Constrained axis 1 is the sampling season. Symbols of samples from early summer (June) are grey, while symbols of samples from late summer (August) are white. For simplicity, only genus names are shown for genera that are represented by a single species.

**Figure 5 insects-14-00782-f005:**
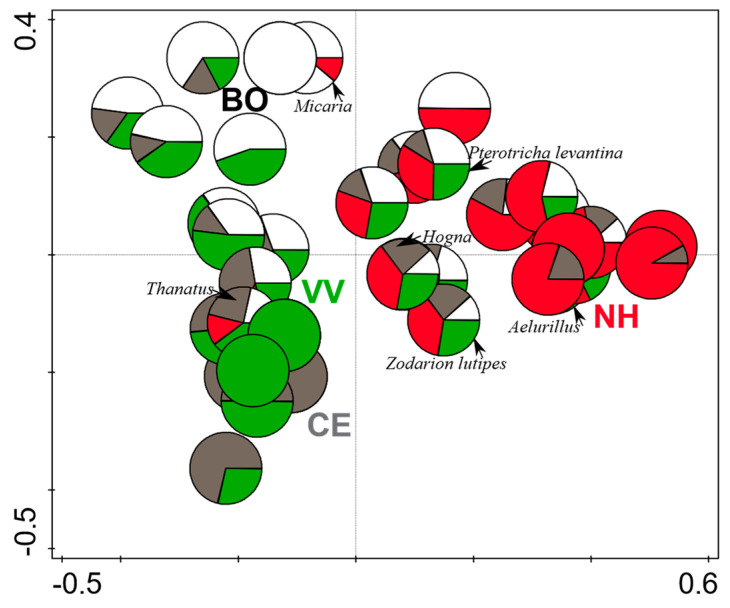
Constrained partial RDA ordination. The distribution of all species across trap positions is represented by species pie diagrams. Constrained axis 1 is the sampling position. “NH” (natural habitat) = red; “BO” (border with the natural habitat) = white; “CE” (center of the vineyard) = grey; and “VV” (focal vineyard in closest to neighboring vineyard) = green. Names of the six most abundant species are shown. For simplicity, only genus names are shown for genera that are represented by a single species.

**Figure 6 insects-14-00782-f006:**
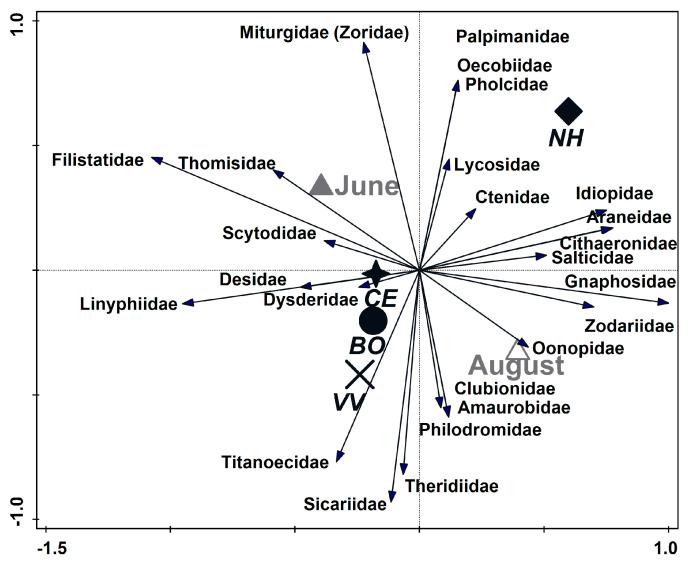
Ordination (RDA) for the 25 families and the significant environmental variables (season and trap position (habitat), first and second axes, respectively), with the site as a co-variable. Each family is represented by an arrow that points in the direction of the steepest increase in abundance. The length of the arrow is a measure of the fit of each family to the ordination axes, namely the strength of multiple correlations between environmental variables and spider families. The distance between the centroids of environmental variables approximates the average dissimilarity between their family values as measured by the Euclidean distance. Diamonds = “NH” (natural habitat); circles = “BO” (border with the natural habitat); stars = “CE” (center of the vineyard); and X = “VV” position (focal vineyard closest to the neighboring vineyard).

**Figure 7 insects-14-00782-f007:**
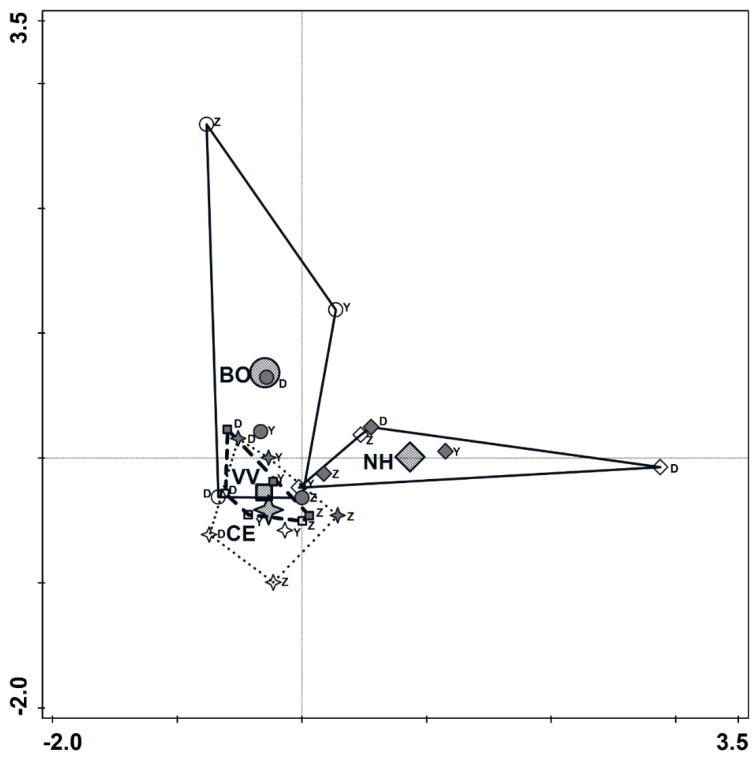
Ordination plot of the 24 samples by their positions on the ordination axes 1, 2. The distance between the samples approximates the average dissimilarity between their species compositions as measured by the Euclidean distance. Symbols of samples from early summer (June) are grey, while symbols of samples from late summer (August) are white. Diamonds = centroid and samples from “NH” (natural habitat); circles = “BO” (border with the natural habitat); stars = “CE” (center of the vineyard); and squares = “VV” (focal vineyard closest to the neighboring vineyard). The letter near each symbol stands for the site (D = Dishon, Y = Yiftah, Z = Zar’it).

**Figure 8 insects-14-00782-f008:**
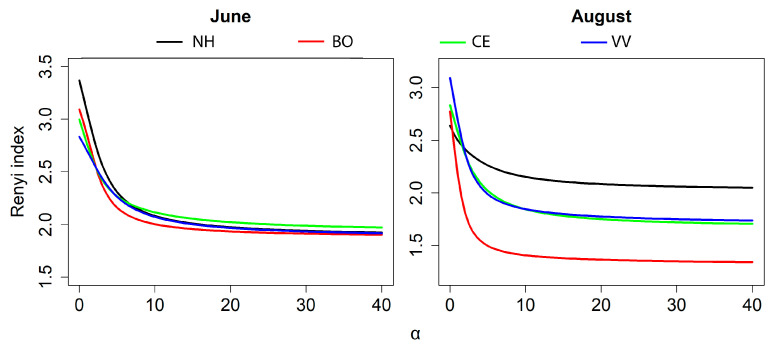
Renyi diversity profiles of the spider assemblages in the four positions within and around vineyards in June (**left**) and August (**right**). “NH” = natural habitat; “BO” = border with the natural habitat; “CE” = center of the vineyard; “VV” = focal vineyard closest to the neighboring vineyard.

## Data Availability

The data presented in this study are available in Appendix A and Appendix B.

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
