# Peer review of "Effects of Natural Habitat and Season on Cursorial Spider Assemblages in Mediterranean Vineyards"

_insects, 2023, doi:10.3390/insects14100782_

Round 1

Reviewer 1 Report

This is a well-written manuscript examining the spider species richness, diversity, as well as evaluating the composition of spider assemblages in vineyards and natural habitats. The core of the work relies on the idea that natural habitats are principal ecosystems that must be preserved near crops. This is because natural habitats are much more diverse in plant species and therefore can maintain more and different animal communities. Animals like spiders are top generalist predators feeding and controlling insect populations that in large number can damage the crops. The study involves studying spider abundance and species composition near and in vineyards to understand the effect of natural habitats considering the time in the season (early and late summer), as different species can be more abundant at different time which ultimately affect the insect community along time. The methods and results are overall nicely presented and the discussion is adequate. This study is a valuable contribution and I only have only minor comments, which I hope can improved the draft. 

Minor comments

Please explain more the experimental design. How distances were selected? Why 100m is the maximum? From introduction it seems that the distances were calculated according to spider mobility and walking distances, but there are no specific references on how far spiders can actually walk. This information would be beneficial to add in methods section. Additionally, it would be important to discuss that maybe 100m is too little to find differences across samplings and that a larger distance can change the results found here.

It would be also important to clarify why you did not include a natural habitat isolated and without vineyards or other agricultural lands to compared.

What happens with the spider species that are not cursorial and therefore cannot fall in the trap? Do you think the abundance and richness is similar to the ones you studied? I would presume that web spiders are very important in vineyards due to their potential to capture insects at certain plant heights and thus, feeding on flying and jumping insects. In contrast, cursorial spiders are restricted to the low level. Please explain the selection on methods section.

Author Response

Reply: We thank the reviewer for the positive comments and for the suggestions below.

Minor comments

Please explain more the experimental design. How distances were selected? Why 100m is the maximum? From introduction it seems that the distances were calculated according to spider mobility and walking distances, but there are no specific references on how far spiders can actually walk. This information would be beneficial to add in methods section. Additionally, it would be important to discuss that maybe 100m is too little to find differences across samplings and that a larger distance can change the results found here.

Reply: We clarified in the text our rationale: “Distances for treatments NH and CE were chosen to avoid edge effects, while those for treatments BO and VV resulted from our intention to sample the two edge positions, one near the natural habitat and another near an adjacent vineyard.”

It would be also important to clarify why you did not include a natural habitat isolated and without vineyards or other agricultural lands to compared.

Reply: In order to compare these four habitats, they had to be selected within the same landscape to ensure that the same regional species pool was present and that the differences between spider assemblages at the four positions were only due to habitat preferences. Several other ecological factors could otherwise have affected the species pool and would confuse the results. Our experimental design is straightforward and has a clear control (NH).

What happens with the spider species that are not cursorial and therefore cannot fall in the trap? Do you think the abundance and richness is similar to the ones you studied? I would presume that web spiders are very important in vineyards due to their potential to capture insects at certain plant heights and thus, feeding on flying and jumping insects. In contrast, cursorial spiders are restricted to the low level. Please explain the selection on methods section.

Reply: We agree with the reviewer that web-building spider species - as well as other predators in the agro-environment (e.g., carabids, ladybirds) - are also important ecosystem service providers. However, our study focused only on ground-active spiders and we cannot speculate on what happened to the other groups based on our results.

Reviewer 2 Report

The study aimed to assess how seasonal changes and proximity to natural habitats influence spider composition and diversity within vineyard ecosystems. The findings revealed higher spider species richness in the early season (post-flowering) compared to the later season just before harvest. Additionally, habitat type significantly influenced species diversity and richness, with the natural habitat exhibiting greater species diversity and richness in the late season. Notably, the study found no significant differences in species assemblage composition within the vineyard habitats. Given the limited knowledge on spider diversity in vineyards in this region, this study offers valuable insights.

Overall, I find the study well executed. I thoroughly enjoyed reading the manuscript. However, I have a few major comments, particularly regarding the importance of adjacent natural habitats in vineyard ecosystems. Additionally, I have some relatively minor comments to consider.

Main comments:

I find your approach and results good as they effectively highlight the significance of natural habitats in promoting spider diversity. However, I believe that the connection between natural habitats and their potential benefits for pest management deserves more attention in the discussion. I recommend considering the addition of a dedicated paragraph to explore this relationship further. I have also included specific comments below (see specific comments).

In your study, you sampled three different sites and four distinct habitat types. It would be beneficial to clarify if you accounted for or examined the differences between these sites, especially considering variations in agrochemical applications. Given the potential site-specific effects on your results, it's important to explore this aspect further. For example, in Figure 1, the central site appears to be surrounded by vineyards on most sides, while the northern site seems less enclosed by vineyards. These differences may influence your findings. Therefore, I suggest discussing the potential influence of agrochemical applications on species composition, as they likely contribute to the reduction in species richness and affect spider assemblage composition. To address this concern, consider adding a section in the methods or results detailing how you controlled for variations in agrochemical applications among the sites, and discuss the implications in the discussion section.

I also have a question regarding the impact of wild boars on your traps. Were the damaged traps concentrated in specific habitats? If so, this could potentially introduce bias into your habitat comparisons. For instance, if 12 out of 18 traps were damaged in a specific habitat, it could complicate the comparison of habitats, potentially underestimating spider species richness in that particular habitat. Please clarify whether the damaged traps were concentrated in specific habitats and outline your approach to addressing this potential bias in your analysis.

Specific comments:

  • Clarify the terminology: could you consistently use either "habitats" or "positions" throughout the manuscript for clarity.

  • Line 49 (l.49): remove “may” for more direct language.

  • l.57 “Fifteen species were found exclusively in the natural habitat”: When mentioning species found exclusively in the natural habitat, consider providing a breakdown of these species (e.g., how many are specialists of vineyards).

  • l.63-64 “Season (early vs. late summer), however, significantly affected spider assemblage composition”: move the sentence about seasonal effects to an earlier section where you discuss the seasonal impact on spider assemblage composition.

  • l.80-87: expand this section to highlight its importance in the discussion.

  • l.88-97: I suggest clarifying the relationship between the natural habitat and vineyards in terms of spider movement and potential benefits for pest control to align better with the introduction/discussion. For example, this paragraph does not align with the last sentence of the introduction “may provide a source of natural enemies for nearby vineyards” (l. 155-156).

  • l. 99-101 “For instance, (…) the warm and cold season [24].” clarrify how the season impacts diversity and abundance – specify whether it has a positive or negative effect.

  • l. 101-103 “At a field scale, (…) assemblage composition” Explain how crop phenology influences assemblage composition.

  • l. 126-127 “management regimes, and regional variables” Here, you show the importance of management regimes and regional variables. How do you account for them in your study? (see one of my main comment)

  • l.174 figure 1: The left part of your figure is missing from the caption. I suppose that the 3 sites are presented on the left part, maybe try another colour to make them more visible, and add their name on the map. I suggest adding labels to the map for each site and improving the visibility of the scale and north arrow.

  • l. 202: “Guide to the Spiders of Britain and Northern Europe”: Are there no identification keys for Mediterranean habitat? I wonder if that would explain why you have morphospecies.

  • l. 215-216 “several variables together” and “environmental variable”: please explain which variables you included in your model. Explain also how you selected your variables in your final model. Did you do a forward (or backward or sterpwise) selection?

  • l. 250-254 figure 2: please increase the font size of your title and text of the axis labels, and of “2a” “2b”

  • l. 256: “Spider richness, composition and diversity”: I propose dividing this section into subsections for spider richness, composition, and diversity for clarity.

  • l. 257-258 “NH, 34 species, of which 15 were found only in this position” Please mention that you found some species only in the vineyard. Especially because you later discuss species “which were associated with the vineyard habitat” (l. 409-410).

  • l. 272-277: consider removing redundant result descriptions in this caption.

  • l. 278-279: mention the percentages of variance explained (as done for the figure 4 lines 278-279).

  • Caption of figure 4 and 5 (l. 299 & 305): expand the caption after “Contrained Partial RDA Ordination”, what was included in each partial RDA?

  • l. 315: suggest replacing “month” with “season” for consistency.

  • l. 366 Figure 7: clarify the type of plot represented in Figure 7.

  • Figures: Suggest using consistent colors for habitat types in all figures to enhance readability.

  • l. 385-390: I would encourage you to elaborate on the potential impact of herbicides and pesticides on spider communities and their prey. For example, discuss the vegetation (which is always the same in the vineyard compared to the NH), or the potential impact of herbicides and pesticides which would kill spiders’ prey and directyl and indirectly impact spiders.

  • l. 407 “such as Micara and Pterotricha”: confirm whether Micara and Pterotricha were found in the samples.

  • l. 407 “migrate” perhaps you shoud not use migrate and disperse interchangeably. Clarify the distinction between "migrate" and "disperse" and use them consistently.

  • l. 420-421: “may be the most suitable habitat” elaborate a bit more, why/how would it be the most suitable habitat?

  • l. 435-437: “This underlines the irreplaceable role of natural habitats to preserve highly diverse spider communities, which in turn may translate into enhanced natural pest control.”: see my main comments. I think you should emphasize the role of natural habitats in preserving diverse spider communities and its implications for enhanced natural pest control in the discussion section.

Author Response

Reply: We thank the reviewer for the positive comments and we appreciate his/her suggestions.

Main comments:

I find your approach and results good as they effectively highlight the significance of natural habitats in promoting spider diversity. However, I believe that the connection between natural habitats and their potential benefits for pest management deserves more attention in the discussion. I recommend considering the addition of a dedicated paragraph to explore this relationship further. I have also included specific comments below (see specific comments).

In your study, you sampled three different sites and four distinct habitat types. It would be beneficial to clarify if you accounted for or examined the differences between these sites, especially considering variations in agrochemical applications. Given the potential site-specific effects on your results, it's important to explore this aspect further. For example, in Figure 1, the central site appears to be surrounded by vineyards on most sides, while the northern site seems less enclosed by vineyards. These differences may influence your findings. Therefore, I suggest discussing the potential influence of agrochemical applications on species composition, as they likely contribute to the reduction in species richness and affect spider assemblage composition. To address this concern, consider adding a section in the methods or results detailing how you controlled for variations in agrochemical applications among the sites, and discuss the implications in the discussion section.

Reply: All sites had one side adjacent to natural habitat and one side adjacent to another vineyard. Our sites were all subject to the same herbicide and insecticide applications. We clarified this aspect in the text: “The vineyards were under integrated pest management (Table S1), all received insecticide and herbicide applications, and were drip-irrigated from June until August (see [38,39] for details).”

I also have a question regarding the impact of wild boars on your traps. Were the damaged traps concentrated in specific habitats? If so, this could potentially introduce bias into your habitat comparisons. For instance, if 12 out of 18 traps were damaged in a specific habitat, it could complicate the comparison of habitats, potentially underestimating spider species richness in that particular habitat. Please clarify whether the damaged traps were concentrated in specific habitats and outline your approach to addressing this potential bias in your analysis.

Reply: We clarified in the text where these losses occurred: “Wild boars damaged 10 traps during the first sampling (2 NH, 4 BO, 4 VV) and 12 traps during the second sampling (8 NH, 1 BO, 3 CE)”. These losses occurred randomly during the first sampling but were mostly concentrated in the NH in the second sampling. However, our conclusions would not have changed as we found that the NH had the highest species richness and a unique spider assemblage. It is possible that our results underestimated the importance of natural habitats.

Specific comments:

Clarify the terminology: could you consistently use either "habitats" or "positions" throughout the manuscript for clarity.

Reply: We use “position” when referring to the specific location of the traps (four positions), each one representing a different habitat. We have revised the text for consistency.

Line 49 (l.49): remove “may” for more direct language.

Reply: Done

l.57 “Fifteen species were found exclusively in the natural habitat”: When mentioning species found exclusively in the natural habitat, consider providing a breakdown of these species (e.g., how many are specialists of vineyards).

Reply: None of the species exclusively found in natural habitats can be defined as a specialist of vineyards.

l.63-64 “Season (early vs. late summer), however, significantly affected spider assemblage composition”: move the sentence about seasonal effects to an earlier section where you discuss the seasonal impact on spider assemblage composition.

Reply: We would like to keep the sentence as it is, as in the previous lines we discuss the results on species richness, while in these lines we discuss the results concerning diversity.

l.80-87: expand this section to highlight its importance in the discussion.

Reply: We expanded this section: “Spiders are important generalist predators in agroecosystems [9]. Worldwide, they are estimated to kill 400-800 million tons of prey every year, much of this likely in crops [10]. Their diverse hunting strategies [11] and ability to establish populations early in the crop season, make them efficient biological control agents of pests [12–14]. Simplification of the agricultural landscape through habitat homogenization has led to a decline in biodiversity worldwide, jeopardizing ecosystem services on which farming systems depend [4]. It is suggested that preserving natural and semi-natural habitats and their vegetation at the habitat and landscape levels will enhance the assemblage composition of spiders, as well as other beneficial groups, and increase their role in biological pest control [15,16]. A diverse spider assemblage is expected to increase their effectiveness as predators in the crop as species-rich assemblage benefit from functional complementarity [17].”

l.88-97: I suggest clarifying the relationship between the natural habitat and vineyards in terms of spider movement and potential benefits for pest control to align better with the introduction/discussion. For example, this paragraph does not align with the last sentence of the introduction “may provide a source of natural enemies for nearby vineyards” (l. 155-156).

Reply: We discuss the role of natural habitats nearby and within vineyards with examples between L111-124: “Despite intensive management, habitats surrounding the vineyards as well as the plant cover within the vineyard, can influence the local spider assemblages. In Mediterranean vineyards in northern Italy, heterogeneous landscapes with nearby woody natural habitats favored the occurrence of ambush hunters, while sheet-web spiders benefitted from less diverse landscapes [31]. Spider densities were positively affected by nearby semi-natural habitats in Germany, but organic vineyards only moderately increased the diversity of spiders compared to conventionally managed ones [32]. In South African vineyards, spider diversity and abundance were enhanced by ground cover within the vineyard [33], by structurally complex natural habitats surrounding the vineyards [34], and by a combination of management regime type and nearby natural habitats [28]. By contrast, there was little effect of the surrounding landscape (woody or grassland) on the abundance of different spider families in Australian vineyards [35], and similarly, of within-vineyard vegetation on spider abundance and species richness [36].”

  1. 99-101 “For instance, (…) the warm and cold season [24].” clarrify how the season impacts diversity and abundance – specify whether it has a positive or negative effect.

Reply: We expanded the text: “Seasonal changes in both the crop and natural habitat can interact with each other affecting spider assemblage composition. This may be due to biotic factors (e.g., changes in the availability of resources) as well as abiotic factors (e.g., thermal tolerances).”

  1. 101-103 “At a field scale, (…) assemblage composition” Explain how crop phenology influences assemblage composition.

Reply: In the sentence (“At a field-scale, the crop phenology and the season affected the activity pattern of different spider taxa [13,25,26] and hence, the assemblage composition”), we offer several citations which, together with the above-mentioned addition (see previous comments) clarify also this aspect.

  1. 126-127 “management regimes, and regional variables” Here, you show the importance of management regimes and regional variables. How do you account for them in your study? (see one of my main comment)

Reply: See reply above. All vineyards received insecticide and herbicide applications.

l.174 figure 1: The left part of your figure is missing from the caption. I suppose that the 3 sites are presented on the left part, maybe try another colour to make them more visible, and add their name on the map. I suggest adding labels to the map for each site and improving the visibility of the scale and north arrow.

Reply: We modified the caption of Figure 1. we enlarge the text and edited the figure

  1. 202: “Guide to the Spiders of Britain and Northern Europe”: Are there no identification keys for Mediterranean habitat? I wonder if that would explain why you have morphospecies.

Reply: There is no key for all spiders of Israel or Mediterranean region, yet identification keys are available for some families, but not all. Many species do not appear in the European keys as they are Near Eastern or African in distribution, and detailed keys are not available. However, we identified most of the species found in our study to a species level if they were adults, only few adults were no identified to species level, in case of families that are not well studied in our region and where the individuals could not identify using other keys or may be an undescribed species. Sub-adults were identified to a genus level. Juveniles were identified to family level, or genus when possible.

  1. 215-216 “several variables together” and “environmental variable”: please explain which variables you included in your model. Explain also how you selected your variables in your final model. Did you do a forward (or backward or sterpwise) selection?

Reply: We rephrased and better explained the method and forward selection in RDA.

  1. 250-254 figure 2: please increase the font size of your title and text of the axis labels, and of “2a” “2b”

Reply: We enlarged the font sizes.

  1. 256: “Spider richness, composition and diversity”: I propose dividing this section into subsections for spider richness, composition, and diversity for clarity.

Reply: Done.

  1. 257-258 “NH, 34 species, of which 15 were found only in this position” Please mention that you found some species only in the vineyard. Especially because you later discuss species “which were associated with the vineyard habitat” (l. 409-410).

Reply: We clarified that four species were exclusively found in the vineyard center: “Eleven species were found in all four positions and only four species were exclusively found in the vineyard center: Pterotricha cambridgei, P. parasyriaca, Ozyptila sp. and one unidentified linyphiid (Figure 3)”.

  1. 272-277: consider removing redundant result descriptions in this caption.

Reply: Done.

  1. 278-279: mention the percentages of variance explained (as done for the figure 4 lines 278-279).

Reply: Here there must have been some confusion, as we indicated the variance explained.

Caption of figure 4 and 5 (l. 299 & 305): expand the caption after “Contrained Partial RDA Ordination”, what was included in each partial RDA?

Reply: In Fig.4, the factor included was sampling seasons (we mentioned “Constrained axis 1 is the sampling season)”. In Fig. 5, the factor was sampling position (i.e., habitat) which we also mentioned: “Constrained axis 1 is the sampling position.”

  1. 315: suggest replacing “month” with “season” for consistency.

Reply: We replace “sampling month” to “sampling season” and “month” to “season” throughout the text.

  1. 366 Figure 7: clarify the type of plot represented in Figure 7.

Reply: Done.

Figures: Suggest using consistent colors for habitat types in all figures to enhance readability.

Reply: the different graph are representing different results, we tried to keep the colors or pattern when possible.

  1. 385-390: I would encourage you to elaborate on the potential impact of herbicides and pesticides on spider communities and their prey. For example, discuss the vegetation (which is always the same in the vineyard compared to the NH), or the potential impact of herbicides and pesticides which would kill spiders’ prey and directyl and indirectly impact spiders.

Reply: We agree with the reviewer that pesticides can have a major negative impact on natural enemies, including spiders. However, as mentioned above, all sites were treated with herbicides and insecticides. Therefore, we do not have a control to discuss whether the spider communities were negatively affected by chemicals (this aspect was not the aim of this study).

  1. 407 “such as Micara and Pterotricha”: confirm whether Micara and Pterotricha were found in the samples.

Reply: We changed the text to: “Nevertheless, some gnaphosid genera, such as Micaria and Pterotricha (which we recorded in this study)…”

  1. 407 “migrate” perhaps you shoud not use migrate and disperse interchangeably. Clarify the distinction between "migrate" and "disperse" and use them consistently.

Reply: We changed “migrate” to “disperse”.

  1. 420-421: “may be the most suitable habitat” elaborate a bit more, why/how would it be the most suitable habitat?

Reply: We expanded on this: “For these species, the ecotone between the vineyard and natural habitat may be the most suitable habitat because it provides resources from both neighboring habitats (see, e.g. [48]). An alternative explanation is that those genera are not related to a particular habitat and use ecotones to disperse in the landscape.”

  1. 435-437: “This underlines the irreplaceable role of natural habitats to preserve highly diverse spider communities, which in turn may translate into enhanced natural pest control.”: see my main comments. I think you should emphasize the role of natural habitats in preserving diverse spider communities and its implications for enhanced natural pest control in the discussion section.

Reply: According to the main comment, we expanded our conclusion on the role of natural habitats in agricultural landscapes: “Our findings agree with results from other studies of spider assemblages in natural and agricultural (both annual and perennial) habitats in other parts of Israel [13,24,48] and indicate that natural habitats neighboring vineyards support spider species that are not found elsewhere in the agricultural landscape. This underlines the irreplaceable role of natural habitats to preserve highly diverse spider communities, which in turn may translate into enhanced natural pest control. Our findings are also in line with recommendations for more sustainable agriculture that emphasizes habitat diversification in agricultural landscapes [Begg et al. 2017]. The establishment of non-crop habitats (e.g. [Albrecht et al. 2020]), as well as the conservation of existing natural and semi-natural habitats (e.g. [Decocq et al. 2016]), can contribute to higher levels of biodiversity and ecosystem services and thus increase the resilience of agricultural systems.”